# Convergent Validation of a Self-Reported Commuting to and from School Diary in Spanish Adolescents

**DOI:** 10.3390/ijerph20010018

**Published:** 2022-12-20

**Authors:** Patricia Gálvez-Fernández, Manuel Herrador-Colmenero, Pablo Campos-Garzón, Daniel Molina-Soberanes, Romina Gisele Saucedo-Araujo, María Jesús Aranda-Balboa, Amador Jesús Lara-Sánchez, Víctor Segura-Jiménez, Pontus Henriksson, Palma Chillón

**Affiliations:** 1“PROmoting FITness and Health through Physical Activity” Research Group, Sport and Health University Research Institute (iMUDS), Department of Physical Education and Sports, Faculty of Sport Sciences, University of Granada, 18071 Granada, Spain; 2La Inmaculada Teacher Training Centre, University of Granada, 18013 Granada, Spain; 3Department of Didactics of Musical, Plastic and Corporal Expression, University of Jaen, 23071 Jaen, Spain; 4UGC Neurotraumatología y Rehabilitación, Hospital Universitario Virgen de las Nieves of Granada, 18014 Granada, Spain; 5Instituto de Investigación Biosanitaria (ibs.GRANADA), 18012 Granada, Spain; 6GALENO Research Group, Department of Physical Education, Faculty of Education Sciences, University of Cádiz, 11519 Cádiz, Spain; 7Research Institute and Biomedical Innovation of Cadiz (INiBICA), 11009 Cádiz, Spain; 8Department of Health, Medicine and Caring Sciences, Linköping University, 581 83 Linköping, Sweden

**Keywords:** active transport, health behaviour, commuting time, students

## Abstract

The aim of this study was to examine the convergent validity of self-reported diary times for commuting to and from school with device-measured positional data (Global Positioning System; GPS) in Spanish adolescents. Methods: Cross-sectional data were obtained from four Spanish public secondary schools in 2021, comprising 47 adolescents and 141 home–school and school–home trips. Participants self-reported the time they left and arrived at home and school through a commuting diary. They wore a GPS device recording the objective time during three trips (i.e., one home–school trip and two school–home trips). Agreement between commuting diary and GPS data regarding home–school trips and school–home trips was evaluated using Bland–Altman plots. Results: Total commuting time differed by 1 min (95% limits of agreement were 16.1 min and −18.1 min) between subjective and objective measures (adolescents reported 0.8 more minutes in home–school trips and 1 more minute in school–home trips compared to objective data). Passive commuters reported 0.7 more minutes and active commuters reported 1.2 more minutes in the total commuting time compared to objective data. Conclusions: Self-reported commuting diaries may be a useful tool to obtain commuting times of adolescents in epidemiological research or when tools to measure objective times are not feasible.

## 1. Introduction

Active commuting to and from school (ACS) is defined as any walking, cycling, running or skateboarding to and from school, and it is linked to several health benefits in youths [1]. This behaviour has been proposed as an opportunity to increase daily physical activity (PA) levels [2,3], and it could occur at least twice (i.e., home–school and school–home) every weekday in all school students, usually from 3 to 18 years old.

In the scientific literature, the majority of studies have used self-reported measures, mainly questionnaires, to assess the mode and frequency of ACS [4] because they are quick and easy to implement [5]. Nevertheless, these questionnaires do not report PA intensity while commuting, nor do they report commuting times. Recently, a systematic review of the benefits of ACS concluded that self-reported ACS might be imprecise, which also may limit possibilities to detect health benefits related to ACS [6]. In addition, the findings of a multi-country study showed that to know commuting characteristics (e.g., time needed, distance covered) is essential to assess the contribution of ACS to overall PA [7]. Thus, objective measurement methods providing information about the duration and intensity of the PA performed during the ACS must be developed and used in children and adolescents [8].

In the scientific literature, there are many ways to measure ACS behaviour. Several studies used self-reported commuting diaries to assess the commuting time along several days [9,10,11,12], which seems to be time-consuming and require hard effort from the respondents [13]. Time intervals are another option to study ACS. Actually, some studies have used theoretical time intervals before the starting school time and after the ending school time, such as 10 min [14], 15 min [15], 30 min [16], 60 min [17] or even 90 min [18], which may not represent the real commuting times. These studies based their decision to use these theoretical time intervals on previous research. Other studies have used objective measures (i.e., Global Positioning System [GPS] and Geographic Information System [GIS]) to identify the time that students needed for commuting to or from school [9,19,20,21]. Nonetheless, GPS has several limitations, such as low signal strength indoors, its relatively expensive price [22] and poor battery life of devices [23]. On the other hand, many studies used accelerometers to provide information regarding PA intensity and the amount of time spent on ACS [9,19,24,25]. In fact, recent studies have recommended a combination of GPS and accelerometers to assess this active behaviour [26,27].

For this reason, studies comparing both objective (using devices such as GPS) and self-reported measures (such as diaries) are required to establish a valid and comparable measurement of the framework times in ACS behaviour. To our knowledge, only one such study [28], which compared self-reported and GPS-measured trip duration in adolescents, has been published. In this previous study based on general trips, all modes of commuting were analysed. Self-reported diary records indicated that trips started and ended later than shown by the GPS data. Moreover, regarding ACS, there are no validation studies about the commuting time in a self-reported diary in the scientific literature. Thus, this study aimed to examine the convergent validity of self-reported commuting diary times to and from school with device-measured positional data (GPS) in Spanish adolescents.

## 2. Methods

### 2.1. Design Study and Procedure

This study analysed cross-sectional data from four public secondary schools across three Spanish cities (Granada, Jaén, and Almería). Data were collected between January and June 2021 using a randomized sampling. These public secondary schools were invited by phone call. Then, the research team conducted an initial meeting with each school’s headmaster and teachers to explain the objectives of the study. After participation approval was obtained from the secondary schools, families signed an informed consent form for the inclusion of their children in the study. This study is part of the PACO Study (“Cycle and Walk to School Study”), and further details can be found elsewhere [29]. The PACO Study has been approved by the Ethics Committee on Human Research (CEIH) at the University of Granada (Reference: 162/CEIH/2016).

Briefly, the adolescents completed a paper-based student questionnaire during the assessment session and a paper-based commuting diary for a week at home. Moreover, they wore a GPS device on the left side of their hip [30] during one and a half school days, recording three trips between home and school (i.e., 1 home–school trip and 2 school–home trips). 

### 2.2. Participants

A total of 169 adolescents from four secondary schools were invited, and 94 adolescents agreed to participate in this study. The inclusion criteria were: (a) to be an adolescent in the 3rd grade of secondary school; (b) to have complete data in a self-reported diary and on a student questionnaire; and (c) to have valid data on a GPS device for three trips (i.e., 1 home–school trip and 2 school–home trips). Of the 94 adolescents, 47 of them met the inclusion criteria. Thus, the final sample size included 47 participants (48% girls) aged 14–15 years old. A total of 141 trips were included (i.e., three trips per participant), comprising 47 home–school trips and 94 school–home trips.

### 2.3. Measures

#### 2.3.1. Measures Sociodemographic Characteristics

Sociodemographic characteristics (i.e., gender, age and full postal address) were self-reported by adolescents through a student questionnaire.

#### 2.3.2. Mode of Commuting to and from School

Adolescents self-reported their usual mode of commuting to and from school during the latest week through the following questions: “How do you usually get to school?” and “How do you usually get home from school?” Both questions have been verified in the Spanish population [11,31]. The possible responses options were: walk, bike, car, motorcycle, school bus, public bus and other. Adolescents were categorized as ‘active commuters’ if they walked or cycled to and/or from school ≥1 trip per day out of 2 daily school trips and ‘passive commuters’ if they commuted to and from school by car, motorcycle, school bus or public bus in both school trips [32].

#### 2.3.3. Self-Reported Commuting Diary

A commuting diary was filled out by adolescents at home for 7 consecutive days, in which they were requested to register: (i) the time they left home, (ii) the time they arrived at school, (iii) the time they left school and (iv) the time they arrived at home (see Appendix A). Adolescents were instructed at school about completing the commuting diary. Three continuous variables (i.e., starting time, ending time and commuting time) were created based on self-reported commuting diary data, hereinafter referred to as self-reported diary time. The self-reported variables of starting time and ending time were based on the exact times of leaving and arriving, respectively, and based on a 24 h clock. The self-reported variable of commuting time was obtained from the subtraction of the starting time variable from the ending time variable. This methodology has also been used by previous studies [33,34].

#### 2.3.4. Device-Measured Positional Data

Positional data were recorded with a GPS device (Qstarz BT-Q1000XT Travel Recorder, International Co., Ltd. Taipei, Taiwan) during one and a half school days (adapting to the schedule of the subject participating in the study) every 15 s (epoch) [14]. Each GPS had a battery to record a maximum of trips: 1 home–school trip and 2 school–home trips. All GPS data were downloaded with Q-Travel, a travel data management software package from Qstarz, and then mapped with Google Earth (Google Inc., Mountain View, CA, USA). Firstly, the locations of both each participant’s home address, self-reported by adolescents, and each participant’s school address were manually geocoded by searching in Google Earth [35]. Secondly, given that there is no standardized protocol for the analysis and interpretation of GPS data [36], the start of a home–school trip was determined as the first GPS point (i.e., first epoch) recorded outside the home, and the end of a home–school trip was defined as the first GPS point (i.e., first epoch) inside the school. In the same way, the start of a school–home trip was determined as the first GPS point (i.e., first epoch) recorded outside the school, and the end of a school–home trip was defined as the first GPS point (i.e., first epoch) inside the home. Thirdly, three continuous variables (i.e., starting time, ending time and commuting time) were created based on device-measured positional data (GPS), hereinafter referred to as objective time. The objective variables of starting time and ending time were based on the exact times of leaving and arriving, respectively, and based on a 24 h clock. The objective variable of commuting time was obtained from the subtraction of the starting time variable from the ending time variable. This methodology has also been used by previous studies [33,34].

#### 2.3.5. Distance

Adolescents self-reported their distance from home to school through the following question: “How far do you live from the school?” The possible response options were: <0.5 km; 0.5 km to <1.5 km; 1.5 km to <3 km; 3 km to <6 km; and ≥6 km. Adolescents were categorized as <1.5 km or ≥1.5 km.

### 2.4. Statistical Analysis

Values are given as means and standard deviations for descriptive analyses. Linear regressions and correlation analyses were conducted as described in Kleinbaum et al. [37]. Agreements between self-reported diary time and objective time for home–school trips and school–home trips were evaluated as described by Bland–Altman [38] plots. The lower and upper limits were calculated as the mean plus/minus 1.96 standard deviations. In the statistical analysis, a value of *p* < 0.05 was considered statistically significant. All analyses were performed using the IBM SPSS Statistics for Windows v25.0 software (SPSS, Inc., Chicago, IL, USA).

## 3. Results

### 3.1. Characteristics of the Trips

The characteristics of the 141 home–school and school–home trips in the self-reported diary data and device-measured positional data (GPS) are described in Table 1. Regarding self-reported diary time, 90% of active commuters spent 30 min or less on home–school trips and 31 min or less on school–home trips. According to the objective time, 90% of active commuters spent 31 min or less on home–school trip and 38 min or less on school–home trips. Moreover, according to self-reported data on the usual mode of commuting to school (home–school trip), 44% used active commuting (i.e., 42% walk and 2% bike), and 56% used passive commuting (i.e., 38% car and 18% public transport). In regard to the self-reported usual mode of commuting from school (school–home trip), 42% used active commuting (i.e., 40% walk and 2% bike), and 58% used passive commuting (i.e., 36% car, 2% motorcycle and 20% public transport). Also, regarding self-reported distance to school (home–school trip), 44% of adolescents lived less than 1.5 km from their school.

### 3.2. Self-Reported Diary Time versus Objective Time

The inter-method agreement between self-reported commuting diary time and objective commuting time in the total commuting time (summing home–school and school–home trips) is displayed in Figure 1. Both self-reported diary time and objective time were significantly correlated (r = 0.598, *p* < 0.001; Figure 1a). Adolescents reported 1 min more in the total commuting time than the objective time, with 95% limits of agreement of 16.1 min and −18.1 min (see Figure 1b). Split by gender, significant correlations were also found (all, *p* < 0.001) (see Appendix A). The same analysis was performed separately for passive (Figure 2) and active (Figure 3) commuters, where significant correlations between self-reported diary time and objective time were found (r = 0.628, *p* < 0.001; Figure 2a, r = 0.469, *p* < 0.001; Figure 3a). For passive commuting, adolescents reported 0.7 min more in the total commuting time than the objective time, with 95% limits of agreement of 19.5 min and −18.1 min (see Figure 2b). For active commuting, adolescents reported 1.2 min more in the total commuting time than the objective time, with 95% limits of agreement of 13.6 min and −11.1 min (see Figure 3b).

The inter-method agreement between self-reported commuting diary time and objective commuting time in the home–school trip is displayed in Figure 4. Both self-reported diary time and objective time were significantly correlated (r = 0.413, *p* < 0.01; Figure 4a). Adolescents reported 0.8 min more in the home–school trip commuting time than the objective time, with 95% limits of agreement of 22.1 min and −20.4 min (see Figure 4b).

The inter-method agreement between self-reported commuting diary time and objective commuting time for the school–home trips is displayed in Figure 5. Both self-reported diary time and objective time were significantly correlated (r = 0.691, *p* < 0.001; Figure 5a). Adolescents reported 1 min more in the school–home trip commuting time than the objective time, with 95% limits of agreement of 15.4 min and −13.4 min (see Figure 5b).

There was inter-method agreement between self-reported diary time and objective time for the home–school and school–home trips regarding the starting and ending time, and they were significantly correlated (all, *p* < 0.001) (see Appendix A).

## 4. Discussion

This study aimed to examine the convergent validity of self-reported commuting diary time with an objective measure (i.e., GPS) in Spanish adolescents using cross-sectional data from 141 trips. Significant correlations between self-reported diary time and objective time in the total commuting time, and in both school–home and home–school trips, were found. Moreover, these correlations between self-reported diary time and objective time were found among passive and active commuters. The results suggest that self-reported commuting diaries may be useful for understanding the commuting times of adolescents. 

Globally, the time intervals in which ACS behaviour occurs before and after school vary in the scientific literature, ranging from 10 min to over 90 min [14,15,16,17,18]. The findings in the current study among adolescents indicate that 90% of active commuters spent about 30 min or less on commuting time during their home–school trip and school–home trip separately, according to the objective time. While previous studies determined the time interval for commuting based on previous literature, the current findings are based on GPS measurements. On the other hand, there is much evidence that home–school distance is the first predictor of ACS in youth [39], and that the distance is highly related with the commuting time (i.e., reported time). Rodríguez-López et al. [40] showed how the threshold distance for walking to school was 1350 m for Spanish adolescents. Because of walking about 2 km per day is roughly equivalent to 30 min of moderate activity [41], perhaps the threshold distance for walking is similar to the 30 min of commuting time found in the current study. Moreover, Spanish adolescents usually live close to schools in the walkable distance buffer [42]. However, despite the fact that all trips to and from school had a maximum duration of 30 min, there are also other time intervals in the current study with different durations (e.g., 8 min, 11 min, 15 min, etc.). Therefore, more studies are required to find a valid and reliable time interval in the scientific literature, in order to be able to establish a cut-off point to assess more accurately the ACS in the future, taking in account the home–school distance.

Providing information on which instrument is more appropriate to assess the commuting time to and/or from school in young people is necessary. The present study found that adolescents reported only around 1 min more in commuting time (i.e., total, starting and ending times) compared with time registered objectively with the GPS device. Our findings are consistent with those previously found in the scientific literature, where self-reported total commuting times were greater in duration than GPS-measured times [43,44]. Concretely, the previous studies showed a difference ranging from 1.2 to 2.2 min more than those measured with a GPS device. With respect to self-reported starting and ending times, a study found about 4 more minutes in self-reported times than in times measured with a GPS device [28]. Therefore, in the current study, adolescents are quite good at reporting their commuting time. Nevertheless, caution is required when making direct comparisons despite generally similar findings, as there are no studies, to our knowledge, dealing with home–school or school–home travel durations measured with self-reported diaries and GPS devices. These results might be explained by several reasons. First, adolescents are previously trained in the correct way to fill in the commuting diary [45]. Second, adolescents have automated starting and ending times to and from school, as they are reported similarly every weekday [46]. In fact, analyses regarding self-reported diary time versus objective time for starting and ending times were performed for home–school and school–home trips. In all commuting times, significant correlations were found (see Appendix A). Third, travelling to and/or from school while accompanied is predictable, and a time routine could be established [47], where every day the active commuters may reach a meeting point at the same time, or passive commuters might have a calculated departure time to make the trip and arrive on time.

In line with the previous results, passive commuters reported their commuting times with more precision than active commuters. Nevertheless, to our knowledge, there are no studies that analyse the correlations between self-reported diary times and objective times by active and passive commuters with which to compare the previous results. This difference could be attributed to passive commuters using the same, direct road route every day, unlike the active commuters, who have greater flexibility to travel down one street or another without traffic signal restrictions. In fact, a study found that direct routes appear to be barriers to children taking up active commuting [48]. Moreover, passive travellers go more or less at the same speed every day (they cannot exceed the speed limit established for each road), so they have calculated the time of the trip in advance (except on occasions when there may be traffic jam). In contrast, active commuters can constantly change their travel speed. Finally, passive commuters have fewer distractions that increase their commuting times during the trip than active commuters. These can be friends, neighbours and/or relatives, shop windows and changes in the environment, among others.

One of the implications of our findings for future studies could be the use of reported measurements (commuting diary) and GPS among adolescents. Moreover, despite adolescents being quite good at reporting their commuting time, it is recommended to use objective measures if feasible. If it is not possible to collect GPS data, a self-reported commuting diary could provide valid and useful information. It is very important that the self-reported diary include the start/end time of each trip from home/school in order to calculate the commuting time. Another implication would be the use of statistical protocols for the analysis and interpretation of GPS data, similar to the ones developed in the present study, to identify the time of start/end from home/school. In addition, for future studies, it could be interesting to ask for the travel time to school in 5 min intervals, in order to have more accuracy regarding ACS. Finally, more studies improving reporting procedures or correlating self-reported commuting diaries with other devices, such as fitness bands or smart watches, are required.

### Study Strengths and Limitations

To our knowledge, this is the first study on the validity of self-reported commuting diary times to and from school using device-measured positional data (GPS) among adolescents. Moreover, the current study analyses starting and ending times and commuting time in both trips (school–home and home–school). The main limitation was that the study was carried out in a region of one country, so the findings cannot be generalized to other regions within the same country or to other countries. Other limitations were that the sociodemographic and distance questions have not been previously validated, and the sample size of the current study was relatively small. Also, further limitations would be the relatively few measured days using GPS and volunteer bias. Finally, the GPS signal could influence detection of an adolescent leaving before or after home/school and the postal addresses self-reported by adolescents, as variations can exist within measurements made by the GPS devices.

## 5. Conclusions

The present study provides a correlation between self-reported diary time and objective time, which means that adolescents are quite good at reporting their commuting time to and from school. For future researchers for whom it is impossible to use objective devices such as GPS, using only self-reported data on commuting to and from school collected in a diary might be appropriate, due to the method’s feasibility. However, these conclusions must be interpreted with caution because the present study was not based on a nationally representative sample of adolescents.

## Figures and Tables

**Figure 1 ijerph-20-00018-f001:**
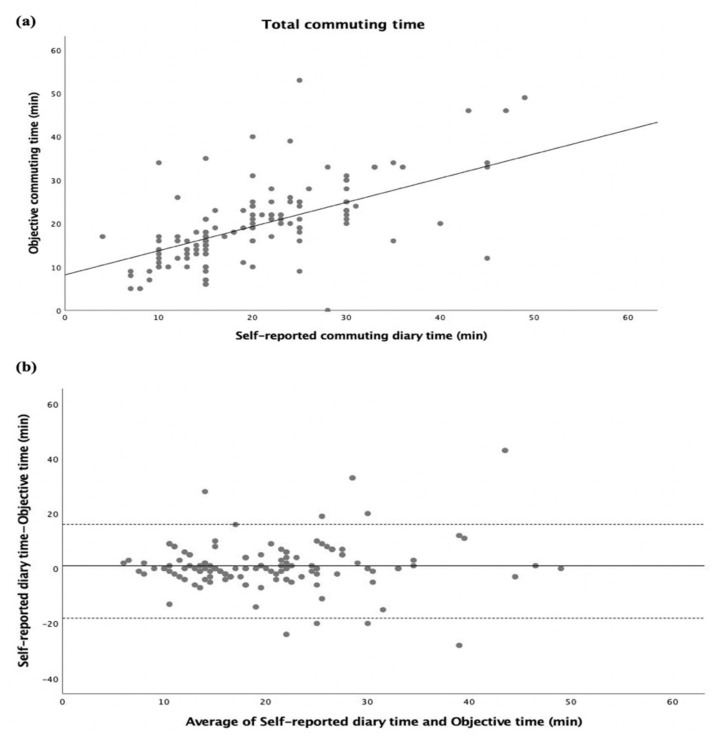
Commuting time expressed as minutes (total commuting time). (**a**) Regression of the objective commuting time (y) vs. (x) the self-reported commuting diary time. The regression equation is y = 8.18 + 0.56x. (**b**) Bland–Altman plot of the total commuting time between self-reported diary time and objective time (y) vs. average of self-reported diary time and objective time (x). The central dotted line represents the mean of differences between the self-reported time measure and the objective time measure; the upper and lower dotted lines represent the upper and lower 95% limits of agreement (mean differences ± 1.96 standard deviations of the differences), respectively.

**Figure 2 ijerph-20-00018-f002:**
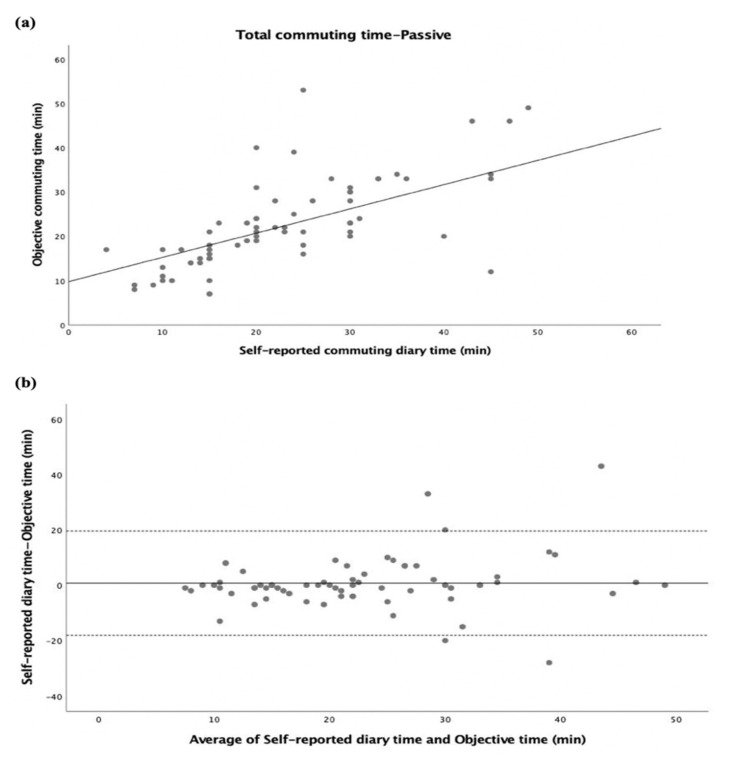
Commuting time expressed as minutes (total commuting time-passive). (**a**) Regression of the objective commuting time (y) vs. (x) the self-reported commuting diary time. The regression equation is y = 9.8 + 0.55x. (**b**) Bland–Altman plot of the total commuting time between self-reported diary time and objective time (y) vs. average of self-reported diary time and objective time (x). The central dotted line represents the mean of the differences between the objective time measure and the self-reported time measure; the upper and lower dotted lines represent the upper and lower 95% limits of agreement (mean differences + 1.96 standard deviations of the differences), respectively. Adolescents reported 3.86 min less in the total commuting time than the objective time (95% limits of agreement were 29.56 min and −21.84 min).

**Figure 3 ijerph-20-00018-f003:**
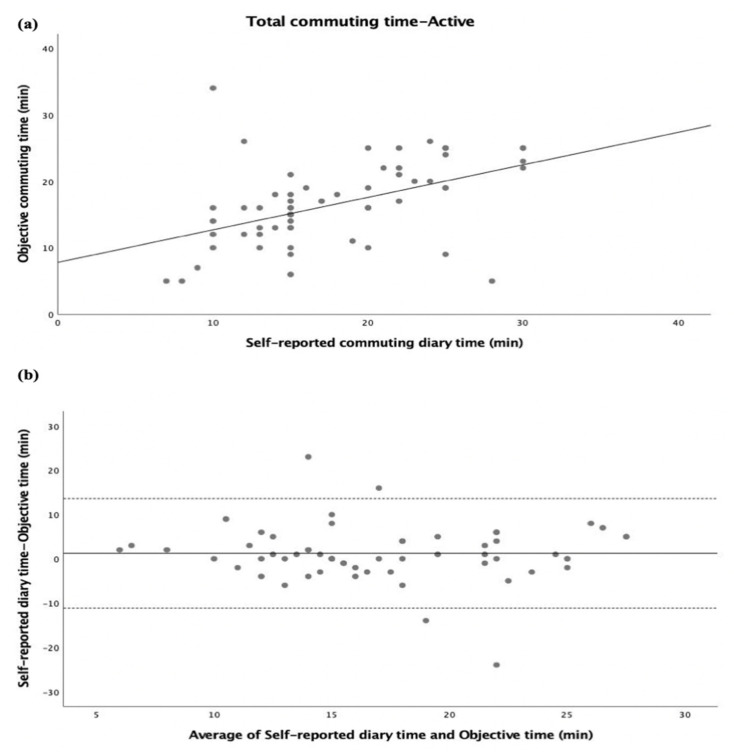
Commuting time expressed as minutes (total commuting time—active). (**a**) Regression of the objective commuting time (y) vs. (x) the self-reported commuting diary time. The regression equation is y = 7.79 + 0.49x. (**b**) Bland–Altman plot of the total commuting time between self-reported diary time and objective time (y) vs. average of self-reported diary time and objective time (x). The central dotted line represents the mean of differences between the objective time measure and the self-report time measure; the upper and lower dotted lines represent the upper and lower 95% limits of agreement (mean differences + 1.96 standard deviations of the differences), respectively. Adolescents reported 6.63 min more in the total commuting time than the objective time (95% limits of agreement were 23.31 min and −36.57 min).

**Figure 4 ijerph-20-00018-f004:**
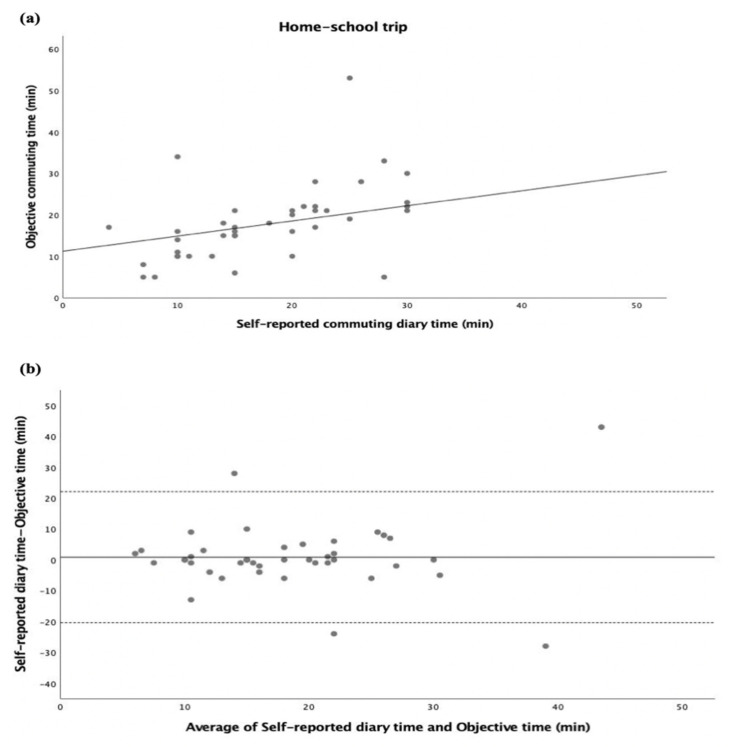
Commuting time expressed as minutes (home–school trip). (**a**) Regression of the objective commuting time (y) vs. (x) the self-reported commuting diary time. The regression equation is y = 11.23 + 0.36x. (**b**) Bland–Altman plot of the commuting time of the home–school trip between self-reported diary time and objective time (y) vs. average of self-reported diary time and objective time (x). The central dotted line represents the mean of differences between the objective time measure and the self-reported time measure; the upper and lower dotted lines represent the upper and lower 95% limits of agreement (mean differences + 1.96 standard deviations of the differences), respectively.

**Figure 5 ijerph-20-00018-f005:**
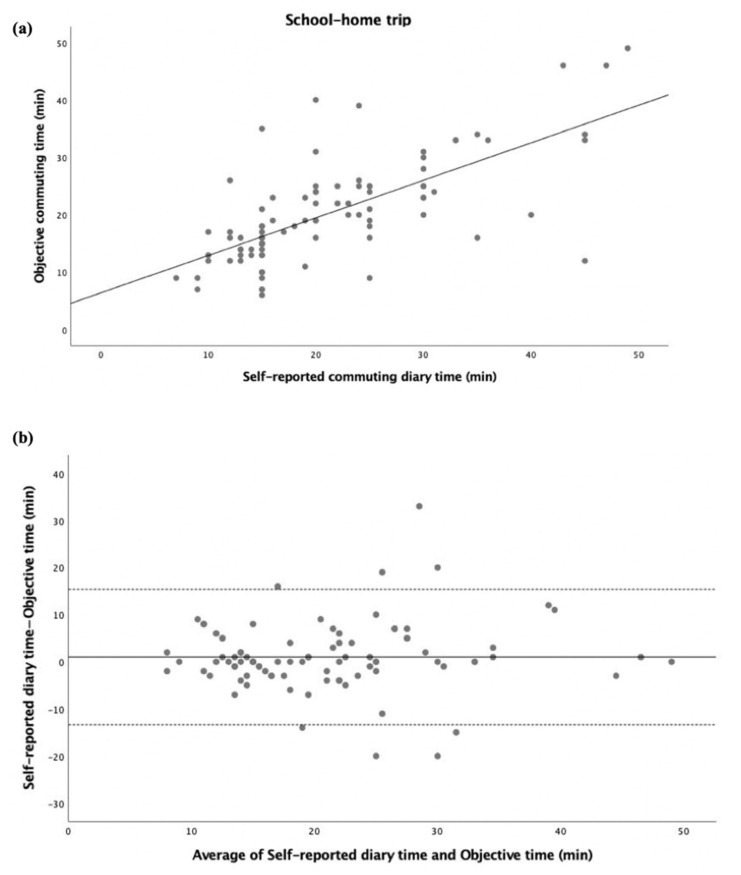
Commuting time expressed as minutes (school–home trip). (**a**) Regression of the objective commuting time (y) vs. (x) the self-reported commuting diary time. The regression equation is y = 6.42 + 0.65x. (**b**) Bland–Altman plot of the commuting time of the school–home trip between self-reported diary time and objective time (y) vs. average of self-reported diary time and objective time (x). The central dotted line represents the mean of differences between the objective time measure and the self-reported time measure; the upper and lower dotted lines represent the upper and lower 95% limits of agreement (mean differences + 1.96 standard deviations of the differences), respectively.

**Table 1 ijerph-20-00018-t001:** Descriptive characteristics of home–school and school–home trips.

Variables	Self-Reported Diary Time N = 47 (X ± SD)		Objective Time N = 47 (X ± SD)		Mean Difference (X ± SD)
			HOME-SCHOOL TRIP		
Starting time *	(7 h 47 min ± 13 min)		(7 h 46 min ± 13 min)		(1 min ± 0 min)
Ending time *	(8 h 6 min ± 9 min)		(8 h 5 min ± 9 min)		(1 min ± 0 min)
Commuting time	(19 min ± 4 min)		(19 min ± 4 min)		(0 min ± 0 min)
Commuting time percentiles					
Percentiles	Active N = 24	Passive N = 23	Active N = 24	Passive N = 23	
Percentile 10	8 min	8 min	5 min	10 min	
Percentile 30	11 min	15 min	10 min	16 min	
Percentile 50	15 min	18 min	16 min	20 min	
Percentile 70	22 min	24 min	21 min	22 min	
Percentile 90	30 min	30 min	31 min	32 min	
Variables	Self-reported Diary time N = 94 (X±SD)		Objective time N = 94 (X±SD)		Mean difference (X±SD)
			SCHOOL-HOME TRIP		
Starting time *	(14 h 40 min ± 9 min)		(14 h 41 min ± 9 min)		(1 min ± 0 min)
Ending time *	(14 h 59 min ± 22 min)		(15 h 1 min ± 11 min)		(2 min ± 11 min)
Commuting time	(19 min ± 13 min)		(20 min ± 2 min)		(1 min ± 11 min)
Commuting time percentiles					
Percentiles	Active N = 43	Passive N = 51	Active N = 43	Passive N = 51	
Percentile 10	12 min	12 min	10 min	10 min	
Percentile 30	15 min	15 min	14 min	17 min	
Percentile 50	15 min	21 min	16 min	22 min	
Percentile 70	20 min	30 min	20 min	29 min	
Percentile 90	31 min	35 min	38 min	35 min	

Notes. X, mean; SD, standard deviation; h, hour; min, minutes; * These are actual times based on a 24-h clock.

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
