# Peer review of "Convergent Validation of a Self-Reported Commuting to and from School Diary in Spanish Adolescents"

_ijerph, 2022, doi:10.3390/ijerph20010018_

Round 1

Reviewer 1 Report

2.3 is on the left side

line 104 remove one space

sociodemographic questionnaire is validated?

From 164 to 187 seems to be doble space

How many use walk, cycle or other? I would like to know each type of active commuting 

And what about the distance?

Small sample size is also a limitation

There is a study regarding active commuting in a representative sample of Spanish children an adolescents. It should be considered in your discussion (doi:10.3390/ijerph17020668) . Discussion need to be more clear and compare with other studies

Could you at least split by gender or consider it in your analysis?

Author Response

Thank you very much for your helpful comments and suggestions. We have addressed all your comments below.

Comment 1

  • 2.3 is on the left side.

 Answer 1

  • Thanks for catching it. We have corrected this mistake. Also, we have reviewed the rest of sections.

 Comment 2

  • line 104 remove one space

Answer 2

  • Thanks for catching it. We have corrected this mistake.

Comment 3

  • Sociodemographic questionnaire is validated?

Answer 3

  • Thanks for your comment. The sociodemographic questions have not been validated. It has been included in study limitations.

Comment 4

  • From 164 to 187 seems to be doble space

Answer 4

  • Thanks for catching it. We have corrected this mistake.

Comment 5

  • How many use walk, cycle or other? I would like to know each type of active commuting

Answer 5

  • Thank you for your suggestion. We have added this information on results section “see characteristics of the trips”.

 Comment 6

  • And what about the distance?

Answer 6

  • Thanks for your comment. The distance had been asked on a self-reported basis but it had not been included in the study. However, now we have included it in method and also at a descriptive level in results. The question about distance has not been validated (included in study limitations).

Comment 7

  • Small sample size is also a limitation

Answer 7

  • Thank you for this point. We totally agree with you. We have included it as limitation. 

Comment 8

  • There is a study regarding active commuting in a representative sample of Spanish children and adolescents. It should be considered in your discussion (doi:10.3390/ijerph17020668) . Discussion need to be more clear and compare with other studies

Answer 8

  • Thank you for your suggestion. The article that you suggest has a very different objective than the objective of our study. Concretely, it examined the associations between the time spent in active commuting and PA levels evaluated by the International Physical Activity Questionnaire for adolescents. Moreover, it has not comparisons between self-reported diary and GPS. In our case, physical activity is not used. Our study is a convergent validation, where the mean goal is to determine if self-reported commuting diary is appropriate for measuring commuting times or not. Nevertheless, attending your comment, we have searched studies again in the scientific literature, to compare our results. We have added two studies to the discussion on total commuting time (see discussion section). However, our study is very novelty in the subject of active commuting. Therefore, to our knowledge, there are no studies that analyze the correlations between self-reported diary time and objective time during the home-school and school-home trips. This is not a problem, it is the beginning for future studies to make comparisons with our study.

Comment 9

  • Could you at least split by gender or consider it in your analysis?

Answer 9

  • Thank you for this comment. We have been considering your comment, and due to the large number of figures in the document, as well as the size of the sample by gender, we have considered including only the analyzes by gender in total commuting time (see Supplementary Material 2).

Reviewer 2 Report

First of all I congratulate the authors on the performed work.

I would have the following observations:

*some of the subjects involved in the study have quite small time travel intervals (less than 15 minutes at least) which do not help in accurately assessing the ACS. Here there is a lot of space for improvement in a future study.

* for self reporting to supply time intervals close to the ones measured by GPS it's always necessary to perform subject training regarding the reporting procedure, which can be subject to different levels of understanding/ misunderstanding when it comes to different ages of the pupils involved.

* being the first study on the comparison between reported times and measured times, at least from my knowledge and also from the authors knowledge, is opens a path towards future improvements both in number of subjects and in improving reporting procedures, or in terms of correlating/obtaining existing on-body GPS device (like fitness bands of smart watches) data.

Author Response

Thank you very much for your helpful suggestions. We have addressed all your comments below.

Comment 1

  • some of the subjects involved in the study have quite small time travel intervals (less than 15 minutes at least) which do not help in accurately assessing the ACS. Here there is a lot of space for improvement in a future study.

Answer 1

  • Thanks for your suggestion. We totally agree with you. The sample is a reflection of the time that adolescents take in the Spanish context, where there is a balanced distribution of public educational schools and it is favoured that they go to the school closest to their home, rewarding the shortest trips. That is to say, the Spanish samples will always be like this. Moreover, based on your comment, in future studies, it would be interesting to ask the travel time to school in 5-minute intervals, in order to have more accuracy in ACS (we have been included it on “implications of our findings for future studies”, see discussion section). For this reason, it is necessary to carry out this type of studies, in order to discriminate your suggestion and temporarily quantify this aspect across the best possible way (self-reported versus objective time).

 Comment 2

  • for self reporting to supply time intervals close to the ones measured by GPS it's always necessary to perform subject training regarding the reporting procedure, which can be subject to different levels of understanding/ misunderstanding when it comes to different ages of the pupils involved.

Answer 2

  • Thank you for your comment. We totally agree with you. In fact, attending to our study, adolescents were instructed about completing the commuting diary at school. We had forgotten to put this information in the method. Now it is included.

Comment 3

  • being the first study on the comparison between reported times and measured times, at least from my knowledge and also from the authors knowledge, is opens a path towards future improvements both in number of subjects and in improving reporting procedures, or in terms of correlating/obtaining existing on-body GPS device (like fitness bands of smart watches) data.

Answer 3

  • Thank you for your comment. We totally agree with you. The current study is only the beginning for future studies about this topic. We have been included it on “implications of our findings for future studies”, see discussion section

Round 2

Reviewer 1 Report

I still observed that you don't add the following reference:

Aparicio-Ugarriza R, Mielgo-Ayuso J, Ruiz E, Ávila JM, Aranceta-Bartrina J, Gil Á, Ortega RM, Serra-Majem L, Varela-Moreiras G, González-Gross M. Active Commuting, Physical Activity, and Sedentary Behaviors in Children and Adolescents from Spain: Findings from the ANIBES Study. Int J Environ Res Public Health. 2020 Jan 20;17(2):668. doi: 10.3390/ijerph17020668. PMID: 31968634; PMCID: PMC7014153.